# Current Advances in the Biomedical Applications of Quantum Dots: Promises and Challenges

**DOI:** 10.3390/ijms241612682

**Published:** 2023-08-11

**Authors:** Nhi Le, Kyoungtae Kim

**Affiliations:** Department of Biology, Missouri State University, 901 S National, Springfield, MO 65897, USA; nhi0407@live.missouristate.edu

**Keywords:** quantum dots, biomedical applications, nanomaterials, interactions

## Abstract

Quantum dots (QDs) are a type of nanoparticle with exceptional photobleaching-resistant fluorescence. They are highly sought after for their potential use in various optical-based biomedical applications. However, there are still concerns regarding the use of quantum dots. As such, much effort has been invested into understanding the mechanisms behind the behaviors of QDs, so as to develop safer and more biocompatible quantum dots. In this mini-review, we provide an update on the recent advancements regarding the use of QDs in various biomedical applications. In addition, we also discuss# the current challenges and limitations in the use of QDs and propose a few areas of interest for future research.

## 1. Introduction

Ever since their discovery in 1981, quantum dots (QDs) have intrigued scientific minds worldwide with their unique physical and optical properties. Quantum dots are nano-sized semiconductor crystals with a broad emission range [1]. Their emission spectra are closely related to their diameter [2,3,4,5,6,7]. This specific characteristic of quantum dots allows for easy manipulation of QDs’ size to achieve the desired fluorescence color during production. Quantum dots are also superior to many other fluorescence probes, such as organic dyes, since QDs’ fluorescence is known to be photobleaching-resistant [8,9,10,11,12]. Due to their physical properties, the structure of quantum dots is relatively stable. In some recent studies, QDs have been demonstrated to remain intact under various pH levels [13], high UV exposure [12], and exposure to oxidative conditions [14]. A standard structure of QDs is composed of a core—often heavy metals—encapsulated by a protective shell [15,16]. QDs can also be conjugated with different types of ligands, which often dictate QDs’ interaction with their surrounding environment [17,18,19,20,21]. The addition of a protective shell and surface ligands increases the solubility and strengthens the structure of QDs [9,11,15,22,23,24]. The protective shell also greatly reduces the exposure of the heavy metal contents residing in the core of QDs [25,26], considered to be one of the main mechanisms of QDs’ toxicity. Another characteristic of quantum dots is that they can be synthesized using many methods. However, these methods are often categorized into either top-down or bottom-up approaches [27]. In the top-down approach, larger precursors are used to form smaller QDs as products through hydrothermal methods [28], electrochemical methods [29], laser ablation [30], etc. On the other hand, the bottom-up approach uses smaller precursors, such as organic molecules, to build larger QD structures through pyrolysis [31] and heat [32]. Although each approach has its own advantages and disadvantages [33], the variety of methods enables developers to choose the best-suited approach.

Due to the diverse types of available quantum dots, QDs are often categorized by their core composition. One of the most widely used cores for quantum dots is cadmium. Cadmium-based QDs, such as cadmium selenide (CdSe), cadmium telluride (CdTe), and cadmium sulfide (CdS), are of great interest to researchers due to their phenomenally high quantum yield compared to other types of QDs [34,35,36]. Thus, cadmium QDs are an excellent candidate for biomedical applications such as trackable drug delivery and bioimaging. However, the potential for cadmium-based QD usage for these biological purposes is currently hindered due to the potential toxicity of their cadmium core [37,38,39,40,41]. Cadmium is well known to cause many major problems for the environment, as well as for human health [42,43,44,45]. It has been reported that cadmium exposure can lead to serious diseases in adults [46] and cause detrimental developmental impairments in children [47]. The United States Environmental Protection Agency (EPA) has also classified cadmium as a probable carcinogenic agent for humans [48]. As such, the usage of cadmium-based QDs raises concerns for many people due to the possible leakage of cadmium ions. In response, countless studies have been dedicated to studying the impact of QDs on the cellular, molecular, tissue, and even organismal levels [49,50,51,52,53,54,55,56]. In the past decade, many studies have reported the toxicity of cadmium-based QDs, including inducing ROS levels [51,57], triggering apoptosis [57], alternate gene expression profiles [58,59,60], damaging the structure and function of mitochondria [61], negative impact on the reproductive system [62,63], causing neurotoxicity [64], and many other undesirable side effects. Due to the countless negative impacts, cadmium QDs are often deemed to be unideal for in vivo biomedical applications. Thus, researchers have shifted their attention to developing other types of QDs that do not contain cadmium.

Indium-based QDs were among the earliest developed non-cadmium QDs. The recent literature suggests that indium-based QDs are less toxic compared to cadmium QDs. Several studies have found that compared to cadmium-based QDs, indium-based QDs have less impact on cell viability, induce less DNA damage, and are overall not toxic [65]. However, contradictory results have also been reported. Davenport et al., reported a similar toxic effect on cell viability between cadmium selenide/zinc sulfide quantum dots (CdSe/ZnS QDs) and indium phosphide/zinc sulfide quantum dots (InP/ZnS QDs) [66]. Cullen et al., found that InP/ZnS QDs reduced the endpoint optical density of the budding yeast *Saccharomyces cerevisiae*, while CdSe/ZnS QDs only caused a prolonged lag phase but did not greatly impact the final optical density [58]. Therefore, more studies need to be conducted before indium-based QDs could be used as a safe alternative. In addition to indium-based QDs, other types of QDs such as copper-based QDs [67], silver-based QDs [68], and many others have also been developed. However, these quantum dots have also been shown to have a negative impact on cells. For example, with ternary copper indium disulfide/zinc sulfide quantum dots (CulnS_2_/ZnS QDs), a high dose induced a minor inflammatory response in the lymph nodes of mice [69]. Silver quantum dots have been reported to exert toxicity by interfering with plants’ photosynthesis processes [70]. Thus, collectively, these studies suggest that each type of QDs has its own toxicity mechanism that needs to be further investigated. Another alternative to cadmium quantum dots is doped quantum dots. Among these, manganese-doped zinc sulfide dots (Mn: ZnS d-dots) are considered to have the most potential due to their low toxicity [71]. Furthermore, Mn-doped QDs have been shown to possess both fluorescence and magnetic properties, making them an ideal candidate for multimodal imaging [72].

Recently, a group of carbon-based QDs have been developed that are claimed to have little-to-no toxicity in both in vitro and in vivo settings. A study in 2019 tested the toxicity of nitrogen-doped carbon quantum dots (NCQDs) in vitro using several cell lines, including HEK293, HepG2, and HeLa cells. The in vitro assessment showed that carbon QDs did not affect the viability of the tested cell lines, even at the highest concentration of 400 µg/mL [73]. Furthermore, in vivo assessment of carbon QDs in albino mice revealed no observable changes in body weight or the levels of enzymes that are often associated with nanomaterial toxicity [73]. In addition to being non-toxic in biological settings, carbon-based QDs are a “greener” type of QDs, as they are not made from heavy metals like traditional QDs. New studies have also reported the synthesis of carbon dots from biowaste [74], which opens the possibility of recycling biowaste for the production of QDs. In addition to these glowing environmental benefits, carbon-based quantum dots also have excellent quantum yields. Therefore, collectively, carbon-based quantum dots are seen as the most promising candidates for biomedical applications. On the other hand, one of the downsides of carbon-based quantum dots is the lack of thorough knowledge. Compared to other types of QDs, carbon-based QDs are relatively new. Thus, more research is needed to further understand carbon-based QDs.

## 2. Ligands

In addition to using alternative core types, modifications to other QD components, such as the surface ligand, have also been explored to maximize QDs’ biocompatibility and efficacy for in vivo applications. Surface ligands are essential for the interaction between QDs and the surrounding environment [17]. Therefore, understanding the behavior of QDs with different conjugated ligands is the first step toward picking the right ligand type for potential clinical usage. In the pharmaceutical field, one of the key factors for an agent to be considered for biological applications is solubility [75]. Thus, numerous strategies have been developed to improve the solubility of chemical treatments [76,77,78] and drug delivery carriers [79,80,81]. For QDs, solubility issues can be avoided by coating the nanocrystal with hydrophilic surface ligands. For example, Ghani et al., replaced the original hydrophobic tri-octyl phosphine oxide (TOPO) ligands on CdSe/ZnS QDs with different types of water-soluble bisphosphonate (BIP)-based ligands, such as ethylene diphosphonate (EDP), imido diphosphonate (IDP), and methylene diphosphonate (MDP). The results showed that the exchanged ligand improved the water solubility and dispersal of CdSe/ZnS QDs [22]. Furthermore, EDP-, IDP-, and MDP-conjugated CdSe/ZnS QDs were shown to have lower toxicity effects compared to CdSe/ZnS QDs conjugated with TOPO ligands. Additionally, the study found that EDP and MDP were significantly taken up by IGROV-1 ovarian cancer cells compared to IDP-conjugated CdSe/ZnS QDs [22]. The difference in the cellular uptake of QDs with different target ligands showed that certain ligands are more suitable than others for biological applications. It is also worth noting that the ligand-exchange step could be skipped if QDs are synthesized in an aqueous phase [71], making it easier to conjugate the desired ligands on QDs.

In another study, Al-Hajaj et al., found that equally sized CdSe(CdZnS) QDs with different conjugated ligands had distinct modes of entry and were taken up in different quantities. Their data revealed that negatively charged CdSe(CdZnS) QDs-CA (QDs conjugated with cysteamine ligands) entered HEK293 human kidney cells and HepG2 human liver cells at a much higher levels compared to positively charged CdSe(CdZnS) QDs, such as CdSe(CdZnS) QDs-CYS (QDs conjugated with cysteine), CdSe(CdZnS) QDs-MPA (mercaptopropionic acid), and CdSe(CdZnS) QDs-DHLA (dihydrolipoic acid) [21]. Ligand types also had an impact on the rate of CdSe(CdZnS) QDs elimination. CdSe(CdZnS) QDs-CA clearance from HepG2 and HEK293 cells took place in the first 3 h post-treatment, while around 80% of CdSe(CdZnS) QDs-CYS was still retained in HepG2 cells after 6 h post-treatment [21].

In addition to affecting the rate of QDs’ cellular uptake, surface ligand types also influence the interaction of QDs with biological components. In 2023, Yu et al., investigated the interaction and impact of two different ligands on CdSe/ZnS QDs—glutathione (GSH-QDs) and dihydrolipoic acid (DHLA QDs)—on the alpha chymotrypsin (ChT) enzyme. Their data revealed that GSH-QDs weakly inhibited ChT’s catalytic activity, while DHLA-QDs greatly inhibited ChT activity. Both DHLA-QDs and GSH-QDs were able to bind to ChT at a 1-to-8 ratio. However, DHLA-QDs have a greater affinity for ChT compared to GSH-QDs, suggesting that the binding affinity of DHLA-QDs with ChT is one of the key factors for the inhibition of ChT activity. The different ligands also resulted in different binding mechanisms between the QDs and ChT, as DHLA-QDs bound to ChT via hydrophobic interactions, while GSH-QDs bound to ChT through hydrogen bonding and van der Waals forces [20]. Recently, some studies have reported that various types of QDs could also interact with common proteins, such as bovine serum albumin (BSA) [82,83,84]. It has been shown that in the presence of CdSe QDs, BSA could bind and form corona protein complexes [85]. The formation of corona protein complexes in the serum could lead to several issues, such as protein structure alteration, protein aggregation, and protein denaturation. In humans, a similar protein called human serum albumin, which is abundant in human serum, has also been reported to form bind to CdSe/ZnS QDs [86]. Thus, it is possible that the same phenomenon could also take place when QDs are used for in vivo applications in humans. The interaction between QDs with different ligands and other biological materials is consistent with prior reports, where ligands played an essential role in the binding of proteins to nanoclusters [87]. Thus, future research should investigate ligand-dependent interactions between QDs and proteins to develop a modified ligand with minimal unwanted QDs–protein interactions.

Collectively, the studies above highlight the importance of choosing the right surface ligands for QDs. Depending on the aim of the application, ligand choice can enhance the safety and efficacy of QDs by altering the interaction between QDs and biological components.

## 3. QDs as a Labeling Agent

Due to their unique characteristics, QDs are vastly useful as a fluorescence label. Thus, a number of studies have attempted to use QDs as a cell labeling agent (Table 1). Previously, Q-tracker 565 and Q-tracker 655 from the Quantum Dots Corporation (Hayward, CA, USA) were shown to effectively label several hematological cell lines (KG-1, HL-60, and SUDHL-16), as well as cells derived from the bone marrow and umbilical cords, by residing in the intracellular space of cells. The labeling of these cells was shown to last from one to two weeks post-incubation. Furthermore, these QDs were found to remain in cells through four cell division cycles, with decreasing QD fluorescence after each cycle. In cells such as HL-60 cells and umbilical-cord-derived CD34+ cells, QDs were also shown to be retained through cell differentiation [88]. These findings showed that QDs could reside intracellularly and label different hematological cell lines for an extended period of time, providing evidence that QDs could be a useful tool for hematological cell imaging. However, the same study also revealed that intracellular labeling by QDs was seen in all tested cell lines, hinting that the labeling by these QDs is not selective. As such, the same group of researchers attempted to target QDs to specific cells by conjugating QDs with streptavidin (QDs-SA). To target cells that specifically express CD33 on the cell surface, QDs-SA were incubated with biotinylated anti-CD33 antibodies prior to cell treatment. The results showed that QDs-SA incubated with biotinylated anti-CD33 selectively bind to cells that express CD33 (HL-60) [88]. These results indicate that when QDs are not conjugated with a selective ligand, they can be randomly internalized by multiple hematological cell lines, thus providing evidence that the presence of targeting ligands is essential for the specific binding.

Apart from whole-cell labeling, QDs have also been used to label specific organelles’ structures. Traditionally, organic dyes are commonly used as fluoroprobes to visualize different structures of cells. However, problems such as short lifetime and weak signal intensity limit their efficiency in cell imaging. As QDs are well known to have photobleaching-resistant fluorescence, they could also be used as superior labeling agents. In one study, streptavidin-conjugated CdSe/ZnS QDs were used to label the actin cytoskeleton of SK-BR-3 cancer cells. The authors found that QDs-streptavidin were able to clearly label the biotinylated F-actin structure of cancer cells. Furthermore, CdSe/ZnS QDs-streptavidin were also able to label other cell structures, such as the nucleus of SK-BR-3 cancer cells pre-incubated with nuclear antigens and biotinylated anti-human IgG [89]. In the most recent 2023 study, neutravidin-conjugated CuInS_2_/ZnS (CIS/ZnS) QDs and neutravidin-conjugated CdSe/ZnS QDs were used as F-actin labeling agents for super-resolution imaging. Although improvements in labeling density are still needed, both QD types effectively labeled F-actin and significantly improved the resolution compared to conventional fluorescence imaging [90]. In addition to fixed structural imaging, QDs could also be used as probes to study real-time cellular processes. Hatakeyama et al., used Qdot (QD655) in combination with HaloTag technology to study the dynamics of the cytosolic myosin motor protein. In this study, QDs were conjugated with a HaloTag ligand and electroporated into cells, where HaloTag ligand-QDs found the protein of interest (myosin) that was fused with the HaloTag protein (Figure 1). In this way, QDs were able to indirectly bind to myosin and act as a probe to study its intracellular movements and interactions. Using this technology, the authors were able to observe myosin’s movement along the actin filament [91]. In a more recent study by Zhang et al., the combination of QDs and HaloTag proteins was used in a similar manner to study the movement of proteins selectively expressed on the surface of mammalian cells through mammalian display technology. In this case, the HaloTag protein was chosen as the displayed surface protein. QDs conjugated with HaloTag ligands (HTL-QDs) were added to act as a probe to track the movement of the displayed HaloTag protein throughout the experiment. Around 30 min after the temperature shift, the displayed HaloTag protein tagged with HTL-QDs moved from the membrane to the cytoplasm. This indicates that the membrane-displayed protein is not always on the cell membrane but is capable of reentering the cells upon temperature changes. The internalized HLT-QDs were transported by membrane-bound vehicles and located near the nucleus but never entered it. The HLT-QDs signals were eventually either detected in the lysosome or recycled back to the membrane surface [92]. Thus, quantum dots are an excellent fluorescence probe for the labeling and imaging of cellular processes.

In addition to using QDs to study the intracellular dynamics of cells, some researchers have also used QDs to study the communications between cells. Extracellular vesicles (EVs) are secreted by cells such as cancer cells and carry information that helps the communication between cells to promote cancer’s proliferation and invasion [93,94,95]. Due to their associations with cancer cells, researchers have recently become interested in using extracellular vesicles like exosomes as cancer biomarkers [96]. To achieve this, it is essential to understand the behavior and interaction of EVs with other components. Therefore, fluorescence probes such as QDs have been used to label and track the dynamics of EVs in recent years. In 2020, Zhang et al., visualized EVs by conjugating them with fluorescence QDs (QDs-EV). The conjugation of EVs with QDs yielded high-quality fixed and live imaging. Furthermore, the interaction between QDs-EVs and microglial BV-2 cells was also detected around 1 h post-co-incubation [97]. In another study, gold carbon dots (GCDs) were used as a nanoprobe to study exosomes secreted by a human breast cancer cell line (SKBR3). This was achieved by using anti-HER2 as an adaptor for GQDs to target HER2 receptors on SKBR3-derived exosomes. By tracking the QDs-labeled exosomes, Jiang et al., were able to observe the uptake of exosomes by HeLa cells. These QDs-labeled exosomes were eventually detected in the lysosome of HeLa cells [98]. QDs showed vast potential in the labeling of EVs and their behavior in vitro. However, it is unclear whether the same efficacy could be achieved in vivo. In the human body, there are multiple cell types that can secret EVs, giving rise to a diverse range of EVs in the tumor microenvironment [99]. As such, targeting a specific type for in vivo labeling of EVs is exceptionally challenging. However, if successful, much insight could be gained through the observation of heterogeneous EV interactions through multicolor imaging.

QDs have also been used to study the behavior and movement of cancer cells in the body. For instance, Voura et al., used QDs to label cancer cells and studied cancer’s distribution to other organs. In this study, CdSe/ZnS QD-labeled B16F10 cancer cells were injected through the tail vein, and their distribution was tracked over time. CdSe/ZnS QD-labeled cells were found mostly in the lungs. However, a small population of cells was also able to arrive at other organs through the lungs’ capillary networks, with low tumor formation frequency. The same study used CdSe/ZnS QDs with different emission ranges to perform multicolor imaging. Two populations of B16F10 cells were labeled with two different types of CdSe/ZnS QDs (510 green and 570 red) and tracked throughout the body. They found that these two populations tended to colonize and form tumors at the same location in the lungs [100]. This result provides a hint that tumor formation in these areas is not random [100]. The use of QDs to study the migration pathways of cancer cells has indeed provided insights into their metastasis behavior. However, a few limitations exist. First, it is unclear whether QD labeling alters any structures, processes, or behaviors of cancer cells. To overcome this issue, the trafficking path of QDs in cancer cells, as well as the specific interactions with cellular components, should be investigated in detail. Second, cancer cells were labeled in vitro before being injected back into mice via the tail vein. Thus, the conditions were different from the extravasation and metastasis of cancer cells in vivo. As such, injected cancer cells may behave differently from cancer cells existing in native biological conditions. In addition, previous studies have shown that nanoparticles, including quantum dots, could be retained in the liver and kidneys for an extended amount of time. Therefore, the interactions of QDs with vital organs, as well as the QDs’ clearance mechanisms, need to be investigated.

**Table 1 ijms-24-12682-t001:** Key studies cited in the Section 3. The table includes title of the cited paper, first author, summary of the quantum dots used in the cited paper, and the in-text citation number.

Title	Author	Summary	Citation
Quantum Dot Labeling and Tracking of Human Leukemic, Bone Marrow, and Cord Blood Cells	Garon et al.,	Qdot 565 and specific antibodies were linked through streptavidin–biotin interaction.	[88]
Immunofluorescent Labeling of Cancer Marker Her2 and Other Cellular Targets with Semiconductor Quantum Dots	Wu et al.,	CdSe/ZnS QDs conjugated with either IgG antibodies or streptavidin were used to label breast cancer cells through the recognition of the HER2 biomarker	[89]
Compact, Fast Blinking Cd-Free Quantum Dots for Super-Resolution Fluorescence Imaging	Nguyen et al.,	CuInS_2_/ZnS (CIS/ZnS) QDs conjugated with neutravidin were used to track biotinylated actin.	[90]
Live-cell Single-molecule Labeling and Analysis of Myosin Motors with Quantum Dots	Hatakeyama et al.,	The target protein was fused with HaloTag, which is recognized by the HaloTag ligand on Qdot (QD655)	[91]
Quantum Dots Tracking Endocytosis and Transport of Proteins Displayed by Mammalian Cells	Zhang et al.,	Qdot conjugated with HaloTag ligand, used to track the HaloTag protein displayed on the surface of the cell membrane	[92]
Quantum Dot Labeling and Visualization of Extracellular Vesicles	Zhang et al.,	QDs-PEG-NH_2_ conjugated to the surface of extracellular vesicles using click chemistry	[97]
Gold-carbon Dots for the Intracellular Imaging of Cancer-derived Exosomes	Jiang et al.,	Gold-based carbon quantum dots conjugated with tumor-specific antibodies to label cancer-derived exosomes	[98]
Tracking Metastatic Tumor Cell Extravasation with Quantum Dot Nanocrystals and Fluorescence Emission-scanning Microscopy	Voura et al.,	CdSe/ZnS QDs were loaded into B16F10 melanoma cells and then injected into mice via the tail vein to track the extravasation of tumor cells	[100]

## 4. Cancer Diagnosis

Cancer is one of the leading causes of death worldwide [101]. Thus, the search for an efficient cancer detection method is urgently needed for cancer diagnostics. Due to their properties, QDs are considered to be potential candidates for cancer detection. Therefore, much research effort has been invested to develop a safe and effective QDs-based cancer detector (Table 2). For example, in 2020, Freitas et al., used CdSe/ZnS QDs to develop electrochemical immunosensors that could recognize the extracellular domain of the human epidermal growth factor receptor 2 (HER2-ECD), a biomarker of breast cancer cells. In this method, the biomarker HER2-ECD was isolated by immobilized antibodies. Afterward, another set of antibodies pre-linked with CdSe/ZnS QDs through streptavidin–biotin interaction were used to label HER2-ECD. Then, a strong acid such as HCl was added to facilitate the release of cadmium ions from CdSe/ZnS QDs, which was then measured by differential pulse anodic stripping voltammetry (DPASV) to quantify the amount of cancer cell biomarkers [102]. Similar methods were also shown to be effective in detecting breast cancer cells and breast-cancer-derived exosomes present in human blood serum [103,104]. Thus, this type of non-invasive, in vitro detection of cancer can be employed for easy early cancer detection.

In addition to measuring the release of ions in quantum dots to quantify cancer cells in blood serum, other strategies using the fluorescence of quantum dots to directly detect cancer cells have also been developed. Due to their low toxicity, graphene quantum dots (GQDs) are among the most potent types of QDs for biological applications [105]. Thus, the use of graphene QDs in cancer detection has caught the attention of researchers. In 2019, differentially doped GQDs, including nitrogen-doped graphene quantum dots (N-GQDs), sulfur-doped graphene quantum dots (S-GQDs), and boron–nitrogen-doped graphene quantum dots (BN-GQDs), were assessed for the labeling of cancer cells (HeLa and MCF-7 cells) vs. normal cells (HEK293). The results showed that when comparing the three types of GQDs, BN-GQDs greatly impacted the cell viability of HeLa cells, while N-GQDs and S-GQDs had no impact on cell viability [106]. Thus, N-GQDs and S-GQDs are more compatible for cell detection purposes. Furthermore, N-GQDs and S-GQDs were found to have pH-induced spectral changes, where blue emission was preferable at neutral pH and green emission was more prominent at acidic pH. As cancer cells and the microenvironment of cancer cells have been reported to be more acidic, the pH-dependent emission changes of N-GQDs and S-GQDs are vastly useful for cancer detection. Indeed, the emission of N-GQDs and S-GQDs was significantly greener in labeled cancer cells (HeLa and MCF-7), while a more prominent blue signal was seen for non-cancerous cells (HEK293) [106]. BN-GQDs, on the other hand, showed no difference between cancerous cells and healthy cells. Thus, it could be inferred that N-GQDs and S-GQDs are useful in detecting cancerous cells in vitro. More investigation regarding the effectiveness of N-GQDs and S-GQDs in in vivo settings is still needed. In addition, factors such as biodistribution and the cytotoxic impacts of N-GQDs and S-GQDs on the major organs and overall health of the model animal should be examined. Finally, it would also be beneficial to study the fate of these GQDs after treatment to determine whether N-GQDs and S-GQDs could safely exit the body. Therefore, much research effort is still needed to determine the potential of N-GQDs and S-GQDs in cancer detection. Other low-toxicity carbon-based QDs such as carbon dots (CDs) have also been used for cancer cell detection. In a 2016 study, gadolinium-doped carbon dots (Gd-CDs) conjugated with folic acid were examined for their potential in dual-modality fluorescence and magnetic resonance imaging of cancer cells. The results showed that a bright fluorescence signal from Gd-CDs was detected from HeLa cells using a confocal microscope. Gd-CDs also showed enhancements in MRI-detectable signals compared to CDs alone. Furthermore, Gd-CDs showed low toxicity towards HeLa cells even with 48 h incubation time at a high treatment concentration of 1 mg/mL [107]. Thus, Gd-CDs are compatible with biomedical applications and should be further investigated for possible applications in in vivo settings. Another study used carbon dots conjugated with folic acid and investigated their ability to recognize cancer cells overexpressing the folic acid receptor (HeLa cells) vs. normal cells (NIH3T3). It was found that after 6 h of incubation, only HeLa cells were brightly fluoresced, while NIH3T3 cells did not fluoresce [108]. This indicates that FA-CDs are effective and can selectively detect HeLa cells. Similarly, Zhang et al., used folic-acid-conjugated carbon dots to distinguish overly expressed folic acid receptors in liver cancer cells (HepG2) vs. normal cells (PC12). They found that FA-CDs exclusively labeled HepG2 [109]. Similar to the previously mentioned GQDs-based cancer detection probes, although FA-CDs were found to be effective in in vitro settings, the next step for FA-CDs would be to test their efficacy in vivo.

With the aim of developing safe and effective quantum dots for cancer detection, a group of researchers used cesium lead bromide quantum dots (CsPbBr_3_) as a scintillator to be detected by X-ray. The CsPbBr_3_ core was double-coated in a silicon dioxide shell (SiO_2_) to limit degradation and prevent toxic core material leakage [110]. The surface of the QDs was also conjugated with CD44 antibodies to specifically target CD44 receptors on pancreatic cells. In an in vitro assessment, QDs were picked up by pancreatic cancer cells (Panc-1), mainly via clathrin-dependent endocytosis. The cesium lead bromide double-silicon-dioxide-encapsulated quantum dots (CPB-SiO_2_@SiO_2_ QDs) were shown to have no impact on cell viability. Furthermore, the in vitro assessment showed that CPB-SiO_2_@SiO_2_ QDs can be clearly detected underneath deep tissues and behind bones using X-ray imaging. Thus, these CPB-SiO_2_@SiO_2_ QDs are safe and effective in the in vitro setting. To examine the relevance of CPB-SiO_2_@SiO_2_ QDs in a biological application, researchers tested their ability to recognize cancer in mice that had transplanted Panc-1 cancer cells. Around 2 h after IV injections, CPB-SiO_2_@SiO_2_ QDs were detected primarily at the tumor site [110]. Dissection of major organs at 2 h after the IV injections revealed high fluorescence signals in the tumor, while minor signals were detected from the spleen and the liver. After 10 days, no signals were detected in the tumor or in any organ, suggesting that the QDs had exited the body by this time. CPB-SiO_2_@SiO_2_ QDs were found in the feces of injected mice 7 h post-injections, and there was no detectible signal after day 7, implying that the QDs had been excreted after 7 days of treatment. No signs of organ defects or changes in body weight in the treated mice were detected [110]. This indicates that these QDs are safe for in vivo cancer detection. One setback of using CPB-SiO_2_@SiO_2_ QDs is the potential leakage of lead (Pb) from the QDs’ core, which may result in various side effects for long-term usage. The same study also showed that although the addition of the double-silicon shell was effective at limiting the exposure of the core’s contents, a minor amount of lead was still leaking from the shelled QDs. Thus, modifications to eliminate this leakage would be beneficial. Additionally, in this study, the signals from CPB-SiO_2_@SiO_2_ QDs were detected by X-ray. The use of X-ray to detect QD probes may potentially risk radiation exposure for patients in long-term use. As such, a QDs-based cancer detection probe that could easily be detected via NIR imaging would be a safer option.

In 2015, a carbon-dot-based cancer detection probe conjugated with a self-guiding molecule Asp (aspartic acid) was developed to target tumors in the brain. One benefit of using CDs-Asp as a cancer detection probe is that their fluorescence signal can be detected in deep tissue. An in vitro assessment of CDs-Asp revealed that CDs-Asp were taken up more by rat glioma cells (C6) compared to L929 cells. In comparison, CDs without Asp conjugated did not show a preference for either cell line [111]. This demonstrates that the addition of Asp was able to increase the selectivity for C6 cells. Another in vivo investigation showed that the signal from CDs-Asp was highest at the glioma tumor site in the brain compared to other brain regions or other organs around 15–30 min after IV injection [111]. This result demonstrates the ability of CDs-Asp to cross through the blood–brain barrier and shows the preference for CDs-Asp towards glioma cancer cells (Figure 2). However, high levels of CDs-Asp were also found in the kidneys and the spleen, along with minor levels in the heart, spleen, and lungs. In addition, the same study also showed a low accumulation of the CDs-Asp in the hippocampus and the cortical layer of the brain [111]. Therefore, although the ability of CDs-Asp is desirable, more improvements need to take place before CDs-Asp could be used as a cancer detection probe. The next important step for CDs-Asp would be to improve the selective targeting of CDs-Asp towards glioma tumors. The goal would be to limit the distribution of CDs-Asp to irrelevant major organs and brain regions. Furthermore, it would also be of great interest to study the retention time of CDs-Asp and their excretion mechanisms. Lastly, the effect of introducing CDs-Asp on the overall health of mice should be investigated.

Recently, one study developed a promising cancer probe using large amino-acid-mimicking carbon quantum dots (LAAM TC-CQDs) with coupled amino and carboxyl ligands. The coupling of these ligands enables a simultaneous interaction of both groups on the ligands with the large neutral amino acid transporter 1 (LAT1) that is highly expressed in cancer cells [112]. In an in vitro test, LAAM TC-CQDs were significantly taken up by the 27 tested cancer cell lines, including HeLa and A549 cells, through LAT1-mediated endocytosis, while they were only minutely taken up by the tested normal cell lines. Additionally, LAAM TC-CQDs were also found to be stable in the pH range from 6 to 8, indicating their stability in the acidic microenvironment of tumors [112]. Furthermore, long-term incubation of LAAM TC-CQDs in the blood serum did not affect their fluorescence intensity or size, thus demonstrating the compatibility and stability of LAAM TC-CQDs for cancer detection. The relevance of LAAM TC-CQDs for biological applications was then tested in an in vivo investigation. It was found that after 10 h post-IV-injection, most of the fluorescence from LAAM TC-CQDs was detected by near-infrared (NIR) imaging at the tumor site for mice transfected with HeLa or A549 cells. Assessment of other large organs revealed minor signals from the lungs, spleen, and kidneys post-treatment. Fascinatingly, IV-injected LAAM TC-CQDs were also found to be able to pass through the blood–brain barrier (BBB) and reach U87 glioma tumors in the brain after 8–12 h. In this case, no other examined large organs showed LAAM TC-CQDs signals. Around 72 h after the IV injection, LAAM TC-CQDs were found to exit the body through urine and feces, indicating that LAAM TC-CQDs can be discarded by the body [112]. Thus, pieces of evidence have suggested that LAAM TC-CQDs are a good candidate for the detection of various tumors. Regardless, some minor challenges still remain for LAAM TC-CQDs. For mice with transplanted HeLa or A549 cancer cells, some LAAM TC-CQD signals were detected in the lungs, liver, and spleen [112]. Thus, investigation regarding the interactions and the long-term effects of LAAM TC-CQDs on these organs is essential. Furthermore, although LAAM TC-CQDs are able to target U87 glioma tumors in the brain, assessments of the impact of LAAM TC-CQDs on nearby brain tissues will also be beneficial to reveal off-target side effects.

**Table 2 ijms-24-12682-t002:** Key studies cited in the Section 4. The table includes title of the cited paper, first author, summary of the quantum dots used in the cited paper, and the in-text citation number.

Title	Author	Summary	Citation
Quantum Dots as Nanolabels for Breast Cancer Biomarker HER2-ECD Analysis in Human Serum	Freitas et al.,	CdSe/ZnS QDs were used to detect HER2-ECD breast cancer cells’ biomarkers	[102]
Immunomagnetic Bead-based Bioassay for the Voltammetric Analysis of the Breast Cancer Biomarker HER2-ECD and Tumor Cells Using Quantum Dots as Detection Labels	Freitas et al.,	CdSe/ZnS QDs linked with antibodies were used to detect the presence of breast cancer cells	[103]
Quantum Dot-based Sensitive Detection of Disease Specific Exosome in Serum	Boriachek et al.,	CdSe QDs modified with streptavidin linked with biotinylated HER-2 or FAM134B antibodies were used to detect the presence of cancer-derived exosomes	[104]
Doped Graphene Quantum Dots for Intracellular Multicolor Imaging and Cancer Detection	Campbell et al.,	Nitrogen-, boron/nitrogen-, or sulfur-doped GQDs synthesized from glucosamine precursors were used to diagnose cancer through pH-sensitive fluorescence response	[106]
Gadolinium-doped Carbon Dots with High Quantum Yield as an Effective Fluorescence and Magnetic Resonance Bimodal Imaging Probe	Yu et al.,	Gadolinium-doped carbon dots (Gd-CDs) were assessed for their biocompatibility and potential to be used in dual-modality fluorescence and magnetic resonance imaging	[107]
Fluorescent Carbon Nanodots Conjugated with Folic Acid for Distinguishing Folate-Receptor-Positive Cancer Cells from Normal Cells	Song et al.,	Carbon nanodots conjugated with folic acid were used to detect cancer cells expressing folic acid receptors.	[108]
Folic Acid-conjugated Green Luminescent Carbon Dots as a Nanoprobe for Identifying Folate Receptor-Positive Cancer Cells	Zhang et al.,	Carbon dots conjugated with folic acid were synthesized from active dry yeast and were used to detect folic-acid-expressing HepG2 cancer cells	[109]
In Vivo Plain X-Ray Imaging of Cancer Using Perovskite Quantum Dot Scintillators	Ryu et al.,	Cesium lead bromide quantum dot scintillators were double-encapsulated in silicon dioxide and conjugated with antibodies against the biomarkers of pancreatic cancer cells. These QDs could be detected even under thick tissues using X-rays	[110]
Self-Targeting Fluorescent Carbon Dots for Diagnosis of Brain Cancer Cells	Zheng et al.,	CD-Asp was synthesized from D-glucose and L-aspartic acid was used to detect and diagnose C6 glioma cells	[111]
Targeted Tumor Theranostics in Mice via Carbon Quantum Dots Structurally Mimicking Large Amino Acids	Li et al.,	LAAM TC-CQDs were synthesized from the precursors 1,4,5,8-tetraminoanthraquinone and citric acid and used for in vivo labeling and detection of HeLa tumors in mice	[112]

## 5. Drug Delivery

In recent years, the use of nanoparticles in drug delivery research has been an increasing trend. Among the different types of nanoparticles, QDs have shown vast potential due to their unique properties. Some in vivo research revealed that upon IV injection, QDs are able to circulate in the body and be retained in major organs like the kidneys and the liver for an extended amount of time [55,113]. The ability of QDs to stay intact when circulating in the body is essential to ensure that QD-derived drug delivery systems are stable until reaching the target site. Simultaneously, QDs’ fluorescence is easily detectible, allowing researchers to clearly map the drug distribution process of QD-derived delivery complexes. Moreover, it was found that QDs are easily taken up by various mammalian cell lines and are targeted to different cellular organelles, including the lysosome [114,115,116,117]. This feature of QDs ensures that QD-derived drug delivery systems are able to enter targeted cells and use the acidic environment in acidic organelles to dissociate therapeutic drugs from the delivery vehicle. For the reasons mentioned above, QDs are a good candidate to be used for drug delivery. On the other hand, problems such as toxicity and target selectivity still need to be resolved to maximize the potential of using QDs as a drug delivery vehicle. In this section, we will introduce some of the recent research on QD-based drug delivery systems (Table 3) and reveal the potential as well as the challenges that come with each type of QDs.

Among the many types of quantum dots, cadmium-based quantum dots are well known for their bright and photobleaching-resistant emission [38]. Thus, this type of QDs could be useful in developing a trackable nanocarrier system. Recently, a CdSe QDs-containing delivery vehicle was developed, where CdSe QDs and the anticancer drug doxorubicin (Dox) were encapsulated in phospholipid micelles. In this complex, CdSe QDs acted as a fluorescence probe to allow for tracking of the drug distribution pathway, while the amphiphilic phospholipid micelle increased the solubility of the complex by confining the hydrophobic CdSe QDs-Dox mixture at the core [118]. This complex was found to effectively carry Dox to HeLa cells. Thus, the CdSe QD-Dox micelles complex has the potential to be used as a drug delivery system. However, a few questions still need to be answered before this complex is ready for usage in drug delivery. First, it is unknown whether this complex could selectively deliver Dox to HeLa cervical cancer cells. Thus, an investigation of the trafficking of CdSe QDs-Dox micelles to a number of cancerous and non-cancerous cell lines could be conducted. In addition, it would be beneficial to perform a study testing the efficacy of CdSe QDs-Dox micelles in killing cancer cells vs. non-cancer cells. Next, this complex could be further tested in an in vivo model. It is well known that the toxicity of cadmium-based materials has been a concern for in vivo applications. Cadmium-based QDs such as CdSe or CdTe QDs are highly toxic due to ion leakage and cadmium content exposure [37,61]. Therefore, more research regarding the CdSe QDs-Dox micelle complex needs to be conducted to determine its efficacy and safety.

Unlike core-only QDs, encapsulated cadmium-based QDs are more promising for biological applications, as the protective shell acts as a barrier to limit the leakage of cadmium ions [15,16]. A study tried using CdSe/CdS/ZnS QDs to carry Dox to rat alveolar macrophage cells. The aim was to maximize the distribution of drugs to specific target cells in the lungs without triggering a sustained inflammatory response. They found that conjugating CdSe/CdS/ZnS QDs to Dox enhanced the levels of Dox delivered to rat alveolar macrophage cells. Furthermore, QDs were found located in the cytoplasm, while Dox was localized in the nucleus [119], indicating an effective release of Dox from the QDs upon entering cells. An in vitro viability assay of alveolar macrophage cells revealed that both free Dox and the QDs-Dox complex reduced the viability of alveolar macrophages. However, at 29 h post-treatment, it appeared that free Dox was much more effective at reducing cell viability compared to the QDs-Dox complex [119]. This phenomenon may be an issue in optimizing treatment at a low dose. On the other hand, the in vivo study revealed that the QDs-Dox complex induced less inflammatory response in the lungs compared to free Dox [119], providing evidence that using QDs as a vehicle to deliver Dox is safer for the lungs compared to the administration of free dox. Even so, it is unclear whether the low inflammatory response of this QDs-Dox complex resulted from the reduction in killing efficacy compared to free Dox, or if this complex has a protective effect against non-target cells. Thus, more research should be conducted. To increase the potential for this complex to be used in drug delivery, the complex should be modified to optimize the killing of target cells while maintaining a minimal impact on non-target lung tissue.

A novel strategy for decreasing off-target effects is to add ligands as homing peptides to guide drug delivery complexes to the targeted cells. One study attempted to develop a drug delivery system that targets prostate cancer cells (PCa) expressing prostate-specific membrane antigen (PSMA) using CdSe/ZnS QDs. In this drug delivery model, CdSe/ZnS QDs were covalently conjugated with an A10 RNA aptamer (Apt, which binds specifically to PSMA), along with the cancer therapeutic agent doxorubicin (Dox). This QD-Apt (Dox) complex was shown to selectively bind to prostate cancer cells that expressed PSMA on the membrane, and it enhanced the toxicity towards PSMA+ prostate cancer cells [120]. Therefore, this complex has the potential to be used as a delivery vehicle. However, issues still need to be resolved for this complex to be utilized in vivo. It was reported that while the QDs-Apt (Dox) complex significantly reduced cell viability in PSMA+ prostate cancer cells, there was also a decrease in cell viability in PSMA− cells [120]. This indicates that there are still some off-target effects in this complex that may induce undesirable side effects. Therefore, an in vivo investigation is needed, as it is unclear how this complex will perform in the normal biological environment and what potential toxicity it may impose to live subjects. Consequently, it is highly recommended that more research efforts must take place before this QDs-APT(Dox) complex could be used in real clinical settings.

In 2020, a study of covalently conjugated quaternary QDs (Ag-In-Zn-S) was conducted with a new generation of chemotherapeutic agents (unsymmetrical bisacridines or UAs) previously shown to have potential effects against lung and prostate cancer cells [121]. According to their in vitro QD stability testing, the QDs-UAs complex was stable in a neutral pH range but underwent disassembly at a low pH [121], suggesting that pH controls the release of UAs once the complex is unstable in an acidic environment in the cell. Interestingly, it was found that the QDs-UAs complex was more readily internalized and caused higher toxicity in the lung cancer cell line (H460) when compared with prostate cancer cells (HCT116). Surprisingly, the complex seemed to have a slight protective effect in normal cell lines (NRC-5 and CCD 841), as the QDs-UAs had a less toxic effect compared to free UAs [121]. Unexpectedly, the same study performed an in vivo investigation using nude mice with HCT116 colon cancer cells and found that the QDs-UAs complex was ineffective in compromising the tumor growth up until 17 days after the treatment with the complex [121]. This might be due to the use of HTC116 cells instead of QDs-UAs-complex-sensitive H460 lung cancer cells, as shown in their in vitro data [121]. Taken together, although this complex possesses vast potential for its protective effects in non-cancerous cells, extreme selective toxicity among the cancer cell lines may lead to economic and practical issues in real lifetimes.

To improve the QDs-UAs complex for cancer drug delivery, in 2022 the same group of researchers used folic acid (FA) ligands as a selective navigator for a QDs-UAs complex, aiming to improve the distribution of the anticancer treatment to cancer cells (Figure 3). It was found that adding FA ligands increased the toxicity of the QDs-UAs complex for all tested cancer cell lines. The addition of FA ligands also enhanced the delivery of chemotherapeutic agents and caused intracellular accumulation of the complex in the tested cancer cell lines (H460, Du-115, and LNCaP) [122]. Thus, modifying the surface ligand expanded the application scope of the QDs-UAs complex to other cell lines than the H460 cell line. However, compared to H460, Du-115, and LNCaP cells, the tested non-cancerous cell lines seemed to be less sensitive but were negatively impacted by the treatment with the QDs-UAs complex with FA ligands [122]. Thus, using FA as a ligand for the QDs-UAs complex seems to somewhat compromise the protective effect towards non-cancerous cells previously reported in 2020. This novel approach of linking quaternary QDs with chemotherapeutic agents could be refined with further modifications of the complex to maximize its effectiveness against various cancer cell lines while minimizing toxic effects on normal cells. Furthermore, a study using nude mice could be beneficial to show the effectiveness of the FA-conjugated QDs-UAs complex in vivo.

Another QDs-based drug delivery system was composed of graphene QDs (GQDs). In this delivery system, GQDs were attached to doxorubicin (Dox) and the selective self-guiding molecule arginine-glycine-aspartic acid (RGD). The results showed that non-conjugated GQDs at a concentration below 100 µg/mL showed no effect on the viability of the tested prostate cancer cell lines (DU-145 and PC-3), while a higher concentration of 400 µg/mL had a minor impact on cell viability [123]. Furthermore, the release of Dox from GQDs has been shown to be pH-dependent, where Dox was released slowly at neutral pH and released rapidly at pH 5 [123]. This pH-dependent feature could be manipulated to modulate the intracellular release of drugs. These results indicate that GQDs have low toxicity and are a good candidate for a well-controlled drug delivery system. Furthermore, it was shown that the addition of the self-guiding ligand RGD peptide increased the toxicity towards the tested cancer cells compared to the non-target carboxylic ligand Dox-GQDs complex [123], providing evidence that the RGD peptide improved the GQDs-Dox complex. It was found that free Dox was toxic towards all tested cell lines; however, it was especially toxic to the non-cancerous cell line MC3T3-EI. Dox-GQDs, on the other hand, nonspecifically lowered the toxicity of the drug to all tested cell lines. With the addition of RGD, the Dox-RGD-GQDs complex was more toxic to cancer cells compared to DOX-GQDs alone. However, this level of toxicity was still significantly lower than that of free Dox. On the bright side, Dox-RGD-GQDs were moderately more effective in killing cancer cells compared to non-cancerous MC3T3-EI cells [123]. Thus, the data suggest that the Dox-RGD-GQDs complex slightly improved the non-selective toxicity of free Dox but, at the same time, decreased the killing efficacy of the drug. Therefore, it is suggested that future research could investigate the following issues: First, the stability of the complex should also be studied to ensure that Dox is strongly bound to the delivery vehicle without detachment prior to entering the targeted cells. This would prevent off-target effects resulting from the leakage of Dox and ensure that the maximum amount of Dox could be delivered to the targeted cells. Furthermore, the ability of cancer and non-cancer cells to uptake the Dox-RGD-GQDs complex should be compared to see whether the RGD peptide really works to selectively guide the delivery complex to the targeted cells. In addition, it would be of great interest to investigate the release of Dox once the complex has entered the cells to ensure complete detachment from the delivery vehicle. Finally, in vivo research should be considered to test the efficacy of this delivery complex in a complete biological setting.

In recent years, a new type of quantum dots called carbon dots have caught the attention of many researchers due to their low toxicity. One group of researchers used fluorescent carbon dots conjugated with hyaluronic acid and carboxymethyl chitosan (CD_C-H_) ligands, which bind to CD44 receptors that are overexpressed in many cancer cells, to make the complex a trackable drug delivery system that effectively delivers Dox to targeted cancer cells. The results from an MTT assay suggested that the (DOX-CD_C-H_) complex is not toxic to NIH3T3fibroblast cells. On the other hand, (DOX-CD_C-H_) showed different degrees of toxicity toward the two breast cancer cell lines (MCF-7 and 4T1) [124]. The viability of the 4T1 cells was significantly more susceptible to this complex compared to MCF-7 cells, which may be due to the higher expression of CD44 in 4T1 cells. The free Dox, on the other hand, was non-selectively toxic to all tested cell lines [124]. Thus, it could be concluded that in an in vitro setting, the (DOX-CD_C-H_) complex selectively kills breast cancer cells that express CD44 while remaining non-toxic to healthy fibroblast cells. Thus, due to its high killing selectivity, this complex is a great candidate for use in drug delivery. The same study also evaluated the efficacy of this (DOX-CD_C-H_) complex in an in vivo setting. The results revealed that (DOX-CD_C-H_) caused a higher reduction in tumor volume and tumor weight compared to free Dox, while maintaining a slightly higher body weight [124]. Therefore, (DOX-CD_C-H_) is an effective delivery complex that could be employed to deliver therapeutic drugs to breast cancer. In the future, additional tests using different animal models and conditions could be the next step for this complex. The safety and efficacy of (DOX-CD_C-H_) should be heavily investigated before moving to clinical testing. If successful, similar strategies could be employed to expand the applications of this trackable, self-guiding delivery vehicle to other cancer types and diseases. Another study synthesized carbon-based fluorescent graphene nano-biochar (NBC) that could be tagged with several types of targeting ligands for selective delivery of the anticancer treatment DHF (5,5-dimethyl-6a-phenyl-3-(trimethylsilyl)-6,6a-dihydrofuro[3,2-b] furan-2(5H)-one) to cancer cells. It was shown that the NBC-based drug delivery system was able to increase the solubility of DHF. Furthermore, the addition of targeting ligands such as riboflavin (R) and biotin (B) significantly promoted the uptake of NBC by A549 lung carcinoma epithelial cells [125]. Thus, these characteristics show that NBC-TL is a good drug delivery system for treatments with low solubility.

**Table 3 ijms-24-12682-t003:** Key studies cited in the Section 5. The table includes title of the cited paper, first author, summary of the quantum dots used in the cited paper, and the in-text citation number.

Title	Author	Summary	Citation
In Vitro Evaluation of Theranostic Polymeric Micelles for Imaging and Drug Delivery in Cancer	Kumar et al.,	CdSe QDs and doxorubicin were co-encapsulated in phospholipid-based polymeric micelles to create a new drug delivery vehicle	[118]
Doxorubicin Conjugated Quantum Dots to Target Alveolar Macrophages/Inflammation	Chakravarthy et al.,	Doxorubicin was conjugated to CdSe/CdS/ZnS quantum dots to deliver the anticancer drug to alveolar macrophages	[119]
Quantum Dot-Aptamer Conjugates for Synchronous Cancer Imaging, Therapy, and Sensing of Drug Delivery Based on Bi-Fluorescence Resonance Energy Transfer	Bagalkot et al.,	Doxorubicin was loaded onto QDs doped with an aptamer that recognizes the PSMA protein on prostate cancer cells. This delivery system features an on/off switch using QDs’ fluorescence to indicate the release of Dox from the delivery complex	[120]
New Unsymmetrical Bisacridine Derivatives Noncovalently Attached to Quaternary Quantum Dots Improve Cancer Therapy by Enhancing Cytotoxicity toward Cancer Cells and Protecting Normal Cells	Pilch et al.,	Quaternary Ag-In-Zn-S QDs loaded with an anticancer agent (unsymmetrical bisacridine derivatives) to target lung and colon cancer cells	[121]
Foliate-Targeting Quantum Dots-β-Cyclodextrin Nanocarrier for Efficient Delivery of Unsymmetrical Bisacridines to Lung and Prostate Cancer Cells	Pilch et al.,	QDs-β-CD-FA-C-2028 were conjugated with folic acid to selectively deliver unsymmetrical bisacridine derivatives to cancer cells	[122]
Fluorescent Graphene Quantum Dots as Traceable, pH-Sensitive Drug Delivery Systems	Qiu et al.,	Arginine-glycine-aspartic acid peptides were linked to the carboxyl groups on the surface of the graphene quantum dot (GQD) to create a self-guiding carrier for doxorubicin. This complex targets the RGD receptors on cancer cells	[123]
Easy Synthesis and Characterization of Novel Carbon Dots Using the One-Pot Green Method for Cancer Therapy	Wang et al.,	Carbon dots were modified with hyaluronic acid and carboxymethyl chitosan using the one-step hydrothermal method. Modified carbon dots were then used to deliver doxorubicin to cancer cells	[124]

## 6. Immunological Study

The human immune system is essential for host defense against foreign pathogens. Thus, understanding the human immune system is the key to the prevention and treatment of disease. However, the immune system is composed of complex multilayer interactions that are hard to study. Therefore, the search for advanced technologies that would allow researchers to gain more insight into the immune system is still ongoing. In recent years, a rising number of researchers have begun to use quantum dots as a tool to study the complex interactions of the immune system, due to QDs’ fluorescence labeling ability. In this section of the review, we will discuss some of the most important studies that used QDs as a tool to examine the immune system (Table 4) and its interactions with pathogens.

For example, in a 2022 paper, CdSe/CdZnS QDs were used to study the uptake of antigens by dendritic cells. Dendritic cells play an essential role in the surveillance and uptake of antigens through tight spaces like tight junctions. However, the specific mechanism is unclear. Thus, Jing et al., used CdSe/CdZnS QDs covered with acrylamide-modified polyacrylic acid as a fluorescence artificial antigen (FAA) to study the uptake of antigens by dendritic cells. The uptake of FAAs by dendritic cells revealed a network of elongated membrane protrusion structures that acted as a tunnel to bring FAAs toward the dendritic cells. With CdSe/CdZnS QDs’ fluorescence, they were able to perform single-particle tracking and to visualized and characterize the lengths and widths of these membrane protrusions [126]. Due to the FAAs, they were also able to observe the merging of adjacent membrane protrusions, as well as the dynamin-dependent endocytic uptake of antigens, as characterized by the lack of fluorescence in membrane fibers after dynamin inhibitor treatment. This study demonstrated a novel way to visualize the dynamics of dendritic cells’ antigen uptake in vitro. This seems to be a powerful technology that enables quality cell imaging [126]. Similar techniques could be employed for other in vitro experiments to study the dynamics and interactions of the immune cells. However, the use of QDs in immune cells is more difficult when it comes to in vivo settings.

Sen et al., investigated the possibility of using QDs to label and track dendritic cells (DCs) in mice, as dendritic cells are important components of the immune system that are responsible for antigen presentation. Their results showed that in a bone-marrow-derived cell mixture, DCs expressing the CD11c marker were able to uptake QDs, while smaller monocytes could not. QDs were contained in vesicles located in the cytoplasm and then delivered to the lysosome around 45 min post-incubation [127]. The internalization of QDs by dendritic cells lasted 48 h and yielded no apparent toxic effect, suggesting that QDs may be suitable for use as a tracking probe for dendritic cells. To evaluate the efficacy of QDs in labeling dendritic cells in vivo, QDs were subcutaneously injected into mice. They found that a portion of the QDs were taken up by resident dendritic cells, and uncaptured QDs traveled to the lymph nodes and were also taken up by migrating, non-resident dendritic cells. After analysis, it was found that QDs preferred to label dendritic cells that expressed the CD11c marker, which was consistent with the in vitro observations. They also compared the labeling of dendritic cells by QDs vs. carboxyl-fluorescein succinimidyl ester (CFSE), which is a non-toxic lymphocyte cell dye. They found that QDs effectively labeled CFSE-positive cells. However, the fluorescence of CFSE was only detected around 200 µm deep, whereas QDs’ fluorescence could be detected around 400 µm underneath tissue [127]. Thus, the data suggest that QDs are a more effective labeling agent in vivo compared to CFSE. However, several issues still need to be addressed before QDs could be used as an alternative labeling agent. First, it is still unclear how QDs would be discarded after leaving dendritic cells. Since QDs have been reported to interact with other cell types and proteins, such as BSA, QDs may nonspecifically interact with other cells and proteins in the body, resulting in undesirable side effects. Furthermore, QDs have been shown to accumulate in major organs such as the kidneys and the liver of mice in vivo. Therefore, it would be beneficial to investigate the fate of QDs after they are expelled from dendritic cells. In addition to the abovementioned challenges in using QDs to study the immune system, it is best to keep in mind that QDs themselves could also affect the immune response. An in vitro assessment of the effects of CdSe/ZnS QDs on immune cells such as macrophages and lymphocytes showed that treatment with 1.25 to 2.5 nM CdSe/ZnS QDs caused a weaker response of these cells against lipopolysaccharide (LPS)—a known strong immunostimulant. The same phenomenon was also seen in vivo, where the injection of QDs into mice caused an alteration in lymphocyte subtypes and manifested a weaker response to LPS stimulation [128]. This reduction in immune reaction is problematic because it hints that treatment with QDs causes a weaker immune response, which may increase susceptibility to infection.

In addition to acting as a probe to study the interaction of the immune system, QDs could also be used to investigate the interaction between host cells and pathogens. Each pathogen has its own pathogenesis mechanism. For example, viruses are one type of pathogen with complicated replication cycles that take place inside the cell. Understanding the viral mode of entry and intracellular behavior is one of the key components in developing a treatment for many viral infections [129,130,131]. As such, much experimental research on new viruses focuses on these aspects. However, traditional methods of studying viruses may include many steps and advanced experiments. Therefore, many researchers turn to the possibility of using fluorescence probes such as QDs to track the fate and behavior of viral particles in real time. A study tagged biotinylated *Novirhabdovirus* (IHNV) with QDs-SA to follow the entry process of IHNV into host cells. They established that IHNV enters the host cells via clathrin-mediated endocytosis, and the intracellular transportation of the virus was dependent on the actin cytoskeleton [132]. Furthermore, the virus was found to be localized at the endosome. As viruses have been known to use acidic conditions as a signal to uncoat, the localization of IHNV may hint that a pH-dependent mechanism is used by IHNV to escape from the membrane-bound endosome [132].

In addition to IHVN, the behavior of other viruses has also been investigated. For example, a study tracking the influenza virus (IAV) stands out because it used dual-color, real-time tracking to investigate the uncoating of the viral particle after entering the host cell (Figure 4). In this study, QD625 was conjugated with the viral ribonucleoprotein complexes (QD625-vRNPs) of influenza viruses using the same SA–biotin interaction as in previously mentioned studies. In the same manner, the envelope of influenza was also tagged with QDs emitting a different color (QD525) [133]. It was found that after internalization, viral uncoating took place during cytoplasmic transportation at the perinuclear regions, manifested by the separation of the signals from QD625 and QD525. In addition to tagging the envelope, the team also tagged differently colored QDs to different segments of vRNPs. This triple-color imaging helped determine that the two segments of vRNP were released separately during viral uncoating and were targeted to different locations following late endosomal release. After escaping from the late endosome, less than half of the QDs-vRNPs were unable to completely enter the nucleus, and one-quarter of the QDs-vRNPs were retained in the cytoplasmic region. The data also showed that QD-vRNPs randomly localize in the nucleus [133]. The use of QDs to study the infection stages of viral particles demonstrated the efficacy of using QDs as a fluorescent labeling agent. By tagging differently colored QDs, researchers can visualize a more complete picture of viral intracellular behavior. It is possible that, in the future, QD labeling could advance to allow for all of the stages of viral infection to be visualized simultaneously. This would provide a novel insight that may be overlooked when studying the individual stages alone. For now, the rest of the viral infection cycle, including replication and release from host cells, could be investigated using QDs. Furthermore, the question of whether tagging with QDs slightly modifies the natural viral infection and replication cycle should be addressed. For example, is the inability to obtain a large percentage of QDs-vRNP due to the properties of labeled QDs? Furthermore, would subsequent steps such as the replication of the viral genome and viral assembly be impacted? These aspects will need to be examined to ensure the accuracy of future findings.

**Table 4 ijms-24-12682-t004:** Key studies cited in the Section 6. The table includes title of the cited paper, first author, summary of the quantum dots used in the cited paper, and the in-text citation number.

Title	Author	Summary	Citation
Fluorescent Artificial Antigens Revealed Extended Membrane Networks Utilized by Live Dendritic Cells for Antigen Uptake	Jing et al.,	CdSe/CdZnS QDs coated with polyacrylic acid were used to track the process of antigen uptake by dendritic cells	[126]
Quantum dots for tracking dendritic cells and priming an immune response in vitro and in vivo	Sen et al.,	Streptavidin-conjugated 655 quantum dots were linked with biotinylated ovalbumin to use as an immune stimulant and dendritic cell tracker	[127]
Immunotoxicity assessment of CdSe/ZnS quantum dots in macrophages, lymphocytes and BALB/c mice	Wang et al.,	The impact of CdSe/ZnS QDs on macrophages and lymphocytes was investigated, and the effects of CdSe/ZnS QDs on the immune system were explored	[128]
Clathrin-Mediated Endocytosis in Living Host Cells Visualized through Quantum Dot Labeling of Infectious Hematopoietic Necrosis Virus	Liu et al.,	Streptavidin-modified Qdots were used to label the surface of biotinylated infectious hematopoietic necrosis virus to track its path in host cells	[132]
Real-time dissection of dynamic uncoating of individual influenza viruses	Qin et al.,	Streptavidin-modified QDs were used to label the biotinylated coat and/or the biotinylated viral ribonucleoprotein complex to study viral uncoating	[133]

## 7. Conclusions

QDs are nanoparticles with exceptional photobleaching-resistant fluorescence properties. Their unique optical characteristics combined with stable physical properties (Figure 5) make QDs one of the most promising candidates for various biomedical applications. In this review, we have discussed recent advances in the development of quantum dots for various biomedical applications. Although QDs show great potential, the field of quantum dot technology is still in its infancy. After assessing the current available literature, we propose the following challenges that future research could focus on for quantum dots to be used in biological and clinical settings.

Quantum dots’ toxicity: One of the greatest challenges in using quantum dots for in vivo applications is their potential toxicity (Figure 5). In recent years, researchers have provided various strategies to minimize the toxicity of quantum dots, such as adding a protective shell to limit the exposure of heavy core metals [16], adding ligands for structural stability and solubility [11,22], and developing non-heavy-metal-containing quantum dots [134]. Furthermore, numerous assays have been performed to test the impact of these quantum dots on cell viability and ROS-mediated apoptotic cell death to ensure the safety of quantum dots at the cellular level [39,51,57,135]. The results showed that although the toxicity of quantum dots was mitigated, some minor levels of toxicity were still observed. In addition, many quantum dot toxicity studies have focused on cellular ROS generation and apoptotic cell death. However, an increasing number of studies have indicated the uptake of quantum dots by a broad variety of cell lines [88,114]. Therefore, it is also important to investigate the specific interactions between QDs and cellular components, because these interactions may interfere with the normal biological processes of cells. However, the assessment of quantum dots’ toxicity in cultured cells alone is not enough to determine the safety of quantum dots, as many of the quantum dots’ biomedical applications take place in vivo. Thus, several studies have tested quantum dots’ trafficking inside mice and recorded the locations of quantum dots’ accumulation. The most frequent organs that accumulate quantum dots are the heart, the lungs, the kidneys, the liver, and the brain [112]. These are vital organs that play a significant role in human health. However, there is still a lack of examination regarding the specific interaction between QDs and these organs at the tissue level, as well as the long-term impact of this interaction. In addition to studying organ damage caused by quantum dots, future research should also focus on how the introduction of quantum dots could impact the immune response. As the immune response is a complicated network of interconnected processes, the possibility that QDs may trigger unwanted responses from the immune system must be considered. Furthermore, these in vivo tests are usually conducted on mice. It is well known that different species have distinct responses to foreign objects. Therefore, it would be beneficial to use a variety of animal models to examine quantum dots’ toxicity before moving to human trials.

Quantum dots’ target-specificity issues: Another obstacle that prevents the use of quantum dots in applications such as drug delivery and cell labeling is off-target issues (Figure 5). Thus far, we have mentioned some strategies in this review, including the modification of quantum dots’ ligands to increase target specificity. For cell labeling, a popular strategy is to take advantage of the great binding affinity between biotin and streptavidin. This strategy works wonderfully for cultured cell labeling. However, the application of this strategy for in vivo applications may yield several problems. First, it is difficult to label the target with biotin or streptavidin, as this would require isolation of the target from the body. Furthermore, biotin is a marker that is highly expressed by several cell types in the body. Therefore, streptavidin-conjugated quantum dots may bind to non-target sites. Other modifications, such as conjugating QDs with ligands with strong binding affinity to the receptors that are naturally highly expressed in target cells, have also been explored. These strategies were reported to significantly improve the targeting of QDs. However, because these receptors are not exclusively expressed on the target cells, minor off-target effects are still an issue. Thus, future research could focus on finding novel ways to improve the targeting of quantum dots.

Biodegradability: In recent years, awareness regarding environmental damage caused by human products has been highly acknowledged. In particular, non-biodegradable waste products such as microbeads have been found to be consumed by both land and aquatic animals. The consumption of these non-biodegradable products becomes increasingly problematic as it moves up the food chain, creating health concerns for many higher feeders, including humans. In response, the use of biodegradable products has been adopted worldwide. Traditionally, quantum dots are composed of heavy metals, making them non-biodegradable (Figure 5). It has been reported that the concentrations of QD deposits in some European regions range from 0.17 to 9.4 ng/g of sediment. Although these concentrations are low, it is possible that these QDs are consumed by bottom feeders such as protozoans. This may lead to further complications as QDs move up the food chain through the predation process, eventually leading to negative impacts on animals and humans [136,137]. Therefore, the development of safe and biodegradable QDs could be the next step for QD research. In addition to biodegradable QDs, greener methods of synthesizing nanomaterials would also help to reduce environmental damage by recycling biowaste. For instance, a novel strategy was used to synthesize carbon-based nanomaterials such as fluorescence graphene nano-biochar (NBC) from orange peel. NBC possesses strong fluorescence emission similar to that of graphene quantum dots prepared via other methods, and it can be tagged with a wide range of selective ligands. Thus, NBC prepared via this method is also a suitable candidate for many biomedical applications [125]. Similarly, other organic precursors, such as cellulose, have also been used to develop carbon-based QDs [31]. Thus, these could be the trend for future research.

Clinical phases: The clinical trial is arguably one of the most important steps of drug and therapy development, as it determines whether the new method [31] is safe and effective for practical implementation. However, a successful clinical trial requires many factors, such as commitment, finance, time, experienced professionals, and delicate experimental planning that abides by government guidelines. Furthermore, tested therapies that work in vitro sometimes fail to deliver the same efficacy in clinical trials. Therefore, the rate of successful clinical trials is low. As discussed above, quantum dot research is still in its infancy, and QDs are not ready to be used in biological applications (Figure 5). According to the data from the NIH, only seven clinical trials have been QD-related as of May 2023. Currently, two of these clinical trials have been withdrawn and five are in their early phases. In these clinical trials, QDs have been used for a wide variety of applications, including drug carrier systems for the treatment of cancer and skin diseases (phase 1) [138], improving the vision of patients with advanced retinitis pigmentosa [139], diagnosis of acute myocardial infarction [140], and as a fluorescence detector to detect CD8+ autoreactive T cells to aid with the development of a vaccine for type I diabetes mellitus [141]. The low number of QD-related clinical trials suggests that much more research effort is still needed before QDs could be used in human biomedical applications.

## Figures and Tables

**Figure 1 ijms-24-12682-f001:**
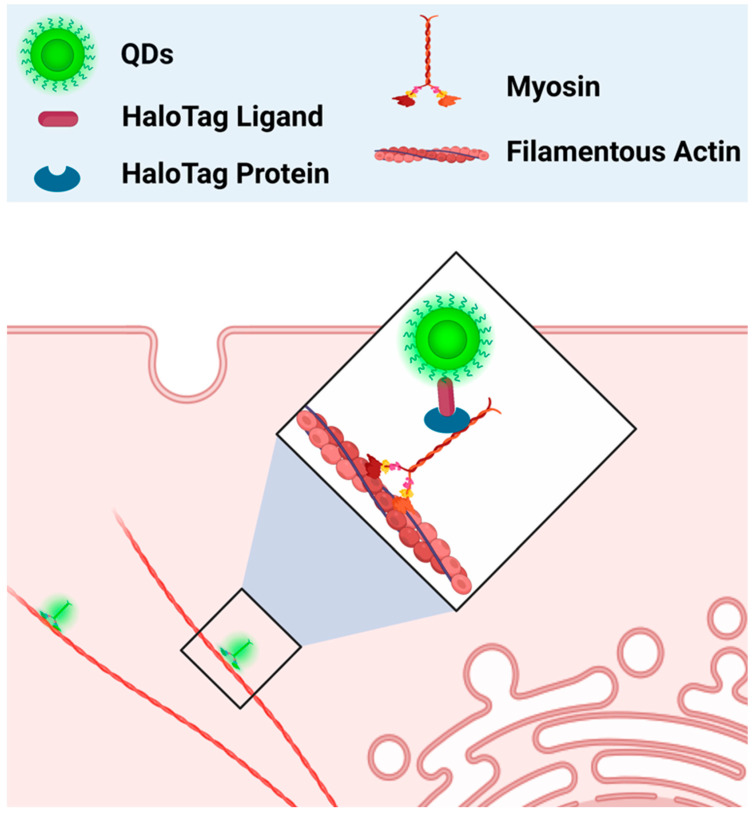
QDs as a fluorescence probe to study the dynamics of myosin’s movement. The HaloTag ligand-QDs complex (HaloTag-QDs) bound to a HaloTag protein that was previously fused with myosin. QDs’ fluorescence helped track the movement of myosin. Diagram created based on findings by Hatakeyama et al. [91].

**Figure 2 ijms-24-12682-f002:**
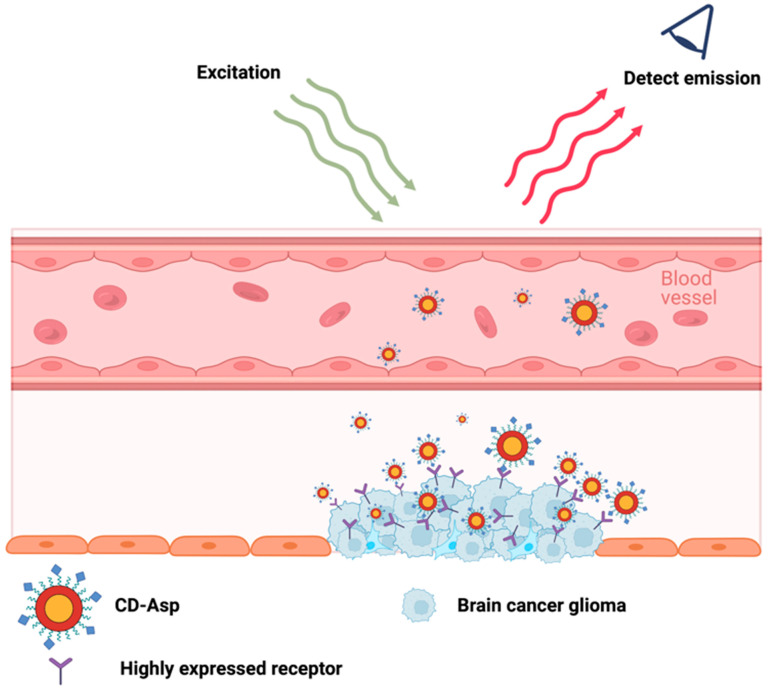
Quantum dots as a cancer detector. Quantum dots with aspartic acid ligands were injected intravenously. QDs-Asp were able to cross the blood–brain barrier and preferably label brain cancer glioma cells. Diagram created based on findings by Zheng et al. [111].

**Figure 3 ijms-24-12682-f003:**
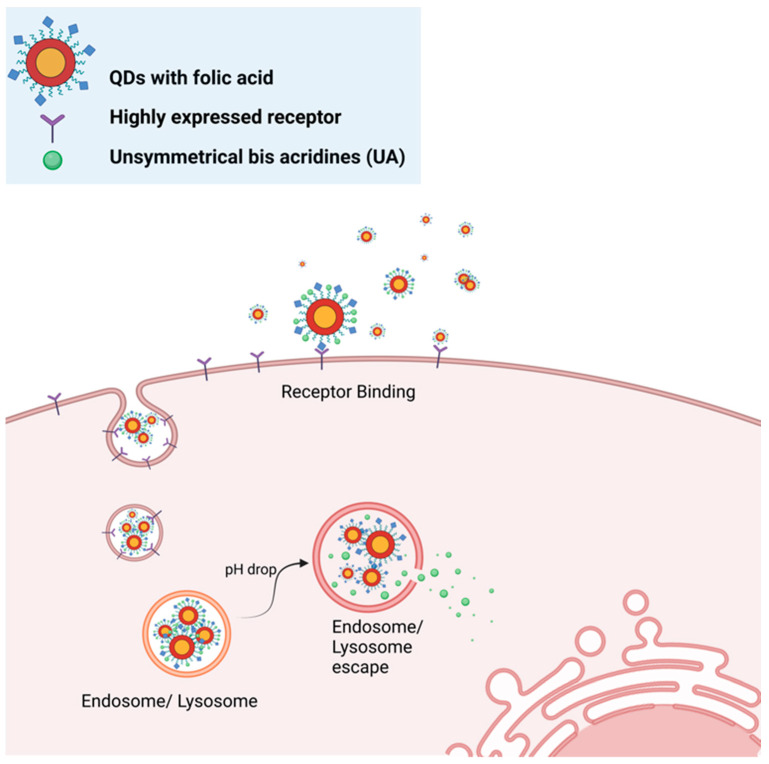
Delivery of anticancer therapeutic agents using a QDs-based drug delivery system: Cancer treatments were loaded on QDs with self-guiding peptides specific to the targeted cancer cells. Upon arrival at the target site, the guiding peptide on the QDs binds to its receptor. The QDs–drug complex is endocytosed and delivered to the endosome/lysosome, where the pH drops, facilitating the detachment of the drug from the QDs and releasing it into the cytoplasm. Diagram created based on findings by Pilch et al. [122].

**Figure 4 ijms-24-12682-f004:**
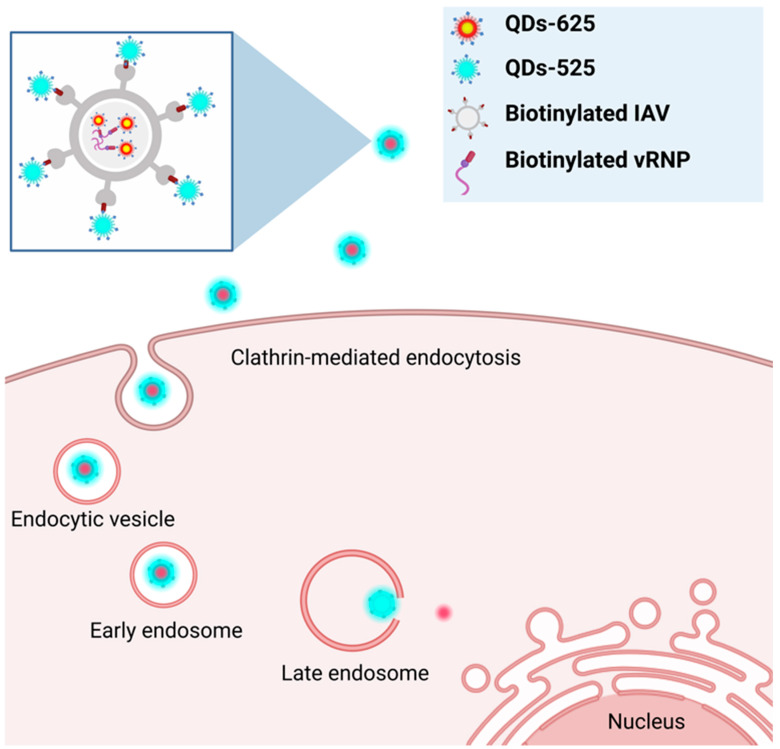
Tracking the uncoating of influenza virus (IAV) by labeling vRNP with QDs-625 (red) and QDs-525 (cyan). The uncoating event was documented by the separation of the two QD signals. Diagram was created based on findings by Liu et al. [132] and Qin et al. [133].

**Figure 5 ijms-24-12682-f005:**
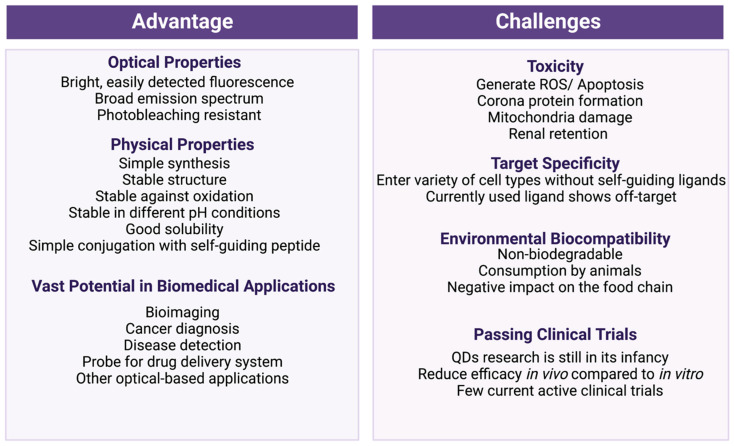
Summary of the advantages and challenges of using QDs in biomedical applications.

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
