# Peer review of "Current Advances in the Biomedical Applications of Quantum Dots: Promises and Challenges"

_ijms, 2023, doi:10.3390/ijms241612682_

Round 1

Reviewer 1 Report

The work is interesting and addresses a relevant and current topic. The detailed discussion allows an approach to the main aspects related to the theme. Minor improvements can be made:

L32 and L54 The author refers to It is thought”… Despite referring to a consideration or idea in the process of being clarified, this expression should be changed to a scientific one. E.g., Recent literature suggests that…”.

L62 In other types of core-types QDs, please avoid repetition.

L99 -  In the acronym "TOPO", the specific designation is missing.

L 147 and L19 Due to their unique physical and optical properties, the information is repeated.

L252 Please correct the word Diagnopsis.

L233 Remove the extra space in QD- labeled cells.

L291 -  Put vs at italic. Also, in L295 and L417 the authors write versus instead of vs. It should be uniformized throughout the paper.

L293 -  In are effective at selectively detect HeLa cells, the authors could rewrite at by and?

L300 -  In With an aim to develop, the authors could rewrite an by the.

L452 In potential to use as please replace by to be used as.

L616- In seen in in vivo tests put in vivo in italics.

Although in section 3. QDs as a labeling agent, the authors refer to the usefulness of QDs applied for the detection of exosomes, HER2 and/or cancer cells, there are some other interesting works reported for diagnosis that can be included:

*Analyst (2017) 142: 22112219, https://doi.org/10.1039/c7an00672a

*Microchimica Acta (2020) 187: 184. https://doi.org/10.1007/s00604-020-4156-4

*Talanta (2020) 208: 120430. https://doi.org/10.1016/j.talanta.2019.120430

Author Response

Please see the attached rebuttal letter.

Best,

Dr. Kim

Reviewer 2 Report

In this review, the authors describe the potential applications of QDs for biomedical applications (labeling, imaging, drug delivery, immunological studies,...). From a general point of view, the manuscript is out interest and well constructed. The following comments should be considered by the authors :

- there are a few repeats, especially for carbon dots. Please revise.

- line 66 : "copper-based QDs" is not appropriate and should be corrected into ternary QDs.

- introduction : the authors should also mention doped QDs that are valuable for biological applications due to their low toxicity (J. Mater. Chem. B 2013, 1, 698-706; ACS Appl. Mater. Interfaces 2018, 10, 34060-34067.).

- paragraph 2 should be slightly revised. The authors should indicate that QDs could also be prepared in aqueous phase to avoid the ligand exchange step.

- line 120 : correct "glutathione".

- the authors must carefully check the whole manuscript and define each abbreviation the first time it is used. See for example CIS (line 178) or indicate CuInS2.

- many of the articles selected by the authors are about core/shell CdSe/ZnS QDs. I recognize that these QDs are attractive for their optical properties but many heavy metal free QDs have emerged in recent years. From my opinion, this should be changed in the manuscript.

- paragraph 4 : the development of bimodal probes (fluorescence and MRI,...) should be described. See for instance, Journal of Alloys and Compounds 2016, 688, 611-619.

only a few minor corrections are required.

Author Response

(The authors gave the same response as above.)

Reviewer 3 Report

The authors report an overview of recent advances in the use of carbon dots for biomedical applications. Even if the reported issues may have relevance in nanomaterials and in biomedical field, I believe that this manuscript can’t be considered for publication in this Journal, since it mainly lacks in novelty. Many recent papers have already reported the biomedical applications of quantum dots (see just few examples: Int J Nanomedicine, 2022, 2, 1951-1970. doi: 10.2147/IJN.S357980; 4open 2023, 6, 1, doi: 10.1051/fopen/2022020; ACS Omega, 2023, 8, 21358-21376. doi: 10.1021/acsomega.2c08183). I would suggest the authors focus their attention on a particular class of quantum dots for a specific application (bioimaging or biosensing or drug delivery, etc.) to make a review on this topic newer and more attractive.

Minor editing of English language required

Author Response

(The authors gave the same response as above.)

Reviewer 4 Report

The manuscript presents a comprehensive review of recent advances in the use of QDs in various biomedical applications. I feel that this manuscript examines an interesting topic, however, in my opinion, the introduction could be improved. Furthermore, I feel that some of the main research work has been left out. 

See my comments below. Once the following points have been properly addressed, the manuscript will be in good condition to be accepted on IJMS.

1. The authors should discuss in a paragraph the main methods used for the preparation of these nanomaterials. In addition, it would be useful to add a table summarising the type of quantum dots, the drug used and the method of synthesis.

2. The works chosen and reviewed for the topic in question must depend on the discretion of the authors. discretion of the authors. I realise that authors may not cite all works related to the topic under review. review. However, I consider works that report relevant techniques or results obtained in recent years for the first time to be relevant, e.g. https://doi.org/10.3390/pharmaceutics14102249;

https://doi.org/10.3390/cancers13091991;

https://doi.org/10.1021/acsomega.8b03369, and so on.

3. Using subscripts in formulae

4. There are many grammatical errors or misuse of words in the manuscript.

Moderate editing of English language required

Author Response

(The authors gave the same response as above.)

Round 2

Reviewer 2 Report

Most of the corrections were made by the authors. The manuscript can be accepted by IJMS.

Author Response

Thank you for your time in reviewing our manuscript.

Dr. Kim

Reviewer 3 Report

The manuscript was greatly improved. However, I believe that this paper can be considered for publication after some revisions. In particular, in the text and in the tables summarizing the reported results, the kind of quantum dots must be reported since their composition play a pivotal role in their application.

Author Response

Thank you for the feedback! We appreciate your help in improving our manuscript.

We attached a rebuttal letter here.

Best,

Dr. Kim

Reviewer 4 Report

Suitable corrections were made. I accept this manunuscript in present form.

Author Response

Thank you for reviewing our manuscript to improve it.

Dr. Kim